# Signatures of Electric Field and Layer Separation Effects on the Spin-Valley Physics of MoSe_2_/WSe_2_ Heterobilayers: From Energy Bands to Dipolar Excitons

**DOI:** 10.3390/nano13071187

**Published:** 2023-03-27

**Authors:** Paulo E. Faria Junior, Jaroslav Fabian

**Affiliations:** Institute for Theoretical Physics, University of Regensburg, 93040 Regensburg, Germany

**Keywords:** valley Zeeman, spin-valley, van der Waals heterostructure, TMDC

## Abstract

Multilayered van der Waals heterostructures based on transition metal dichalcogenides are suitable platforms on which to study interlayer (dipolar) excitons, in which electrons and holes are localized in different layers. Interestingly, these excitonic complexes exhibit pronounced valley Zeeman signatures, but how their spin-valley physics can be further altered due to external parameters—such as electric field and interlayer separation—remains largely unexplored. Here, we perform a systematic analysis of the spin-valley physics in MoSe${}_{2}$/WSe${}_{2}$ heterobilayers under the influence of an external electric field and changes of the interlayer separation. In particular, we analyze the spin ($S_{z}$) and orbital ($L_{z}$) degrees of freedom, and the symmetry properties of the relevant band edges (at K, Q, and Γ points) of high-symmetry stackings at 0° (R-type) and 60° (H-type) angles—the important building blocks present in moiré or atomically reconstructed structures. We reveal distinct hybridization signatures on the spin and the orbital degrees of freedom of low-energy bands, due to the wave function mixing between the layers, which are stacking-dependent, and can be further modified by electric field and interlayer distance variation. We find that H-type stackings favor large changes in the g-factors as a function of the electric field, e.g., from −5 to 3 in the valence bands of the H${}_{h}^{h}$ stacking, because of the opposite orientation of $S_{z}$ and $L_{z}$ of the individual monolayers. For the low-energy dipolar excitons (direct and indirect in *k*-space), we quantify the electric dipole moments and polarizabilities, reflecting the layer delocalization of the constituent bands. Furthermore, our results show that direct dipolar excitons carry a robust valley Zeeman effect nearly independent of the electric field, but tunable by the interlayer distance, which can be rendered experimentally accessible via applied external pressure. For the momentum-indirect dipolar excitons, our symmetry analysis indicates that phonon-mediated optical processes can easily take place. In particular, for the indirect excitons with conduction bands at the Q point for H-type stackings, we find marked variations of the valley Zeeman (∼4) as a function of the electric field, which notably stands out from the other dipolar exciton species. Our analysis suggests that stronger signatures of the coupled spin-valley physics are favored in H-type stackings, which can be experimentally investigated in samples with twist angle close to 60°. In summary, our study provides fundamental microscopic insights into the spin-valley physics of van der Waals heterostructures, which are relevant to understanding the valley Zeeman splitting of dipolar excitonic complexes, and also *intralayer* excitons.

## 1. Introduction

Transition metal dichalcogenides (TMDCs) monolayers host a plethora of fascinating physical properties: for instance, these materials are direct band gap semiconductors with strong spin-orbit coupling (SOC) [1,2,3,4,5], and display robust signatures of excitonic and valley contrasting physics [6,7,8,9,10,11,12,13,14]. Furthermore, the van der Waals nature of TMDC materials facilitates the vertical stacking and integration of different layers, the key ingredient driving the burgeoning research field of van der Waals heterostructures [15,16,17,18]. These novel heterostructures offer a fantastic playground in which to investigate the emergent physical phenomena and effects of proximity interaction, which is certainly not limited to TMDCs [19,20,21,22], but also encompasses other attractive materials, such as graphene [13,23,24,25,26,27], boron nitride [28,29,30], ferromagnetic CrX${}_{3}$ (X = I,Br) [31,32,33,34,35], and many others.

In particular, for the TMDC-based van der Waals heterostructures, one relevant system of great interest is the MoSe${}_{2}$/WSe${}_{2}$ heterobilayer: in this system, the type-II band alignment favors the appearance of interlayer excitons, also known as dipolar excitons, with electrons and holes localized in different layers [36,37,38]. Interestingly, dipolar excitons exhibit long recombination times [39,40,41], robust out-of-plane electric dipole moments [42,43,44,45,46], and giant valley Zeeman signatures that are dependent on the twist angle and atomic registry [20,43,47,48,49,50,51]: these are attractive features that could be exploited in exciton-based opto-spintronics devices [17,18], such as dipolar excitons quantum emitters [43] and diode transistors [46]. Furthermore, these MoSe${}_{2}$/WSe${}_{2}$ are suitable platforms on which to study the optical signatures of moiré superlattices [19,20,52,53] and atomic reconstruction [54,55,56,57]. From a fundamental perspective, there is still some debate about the energy bands that constitute these dipolar excitons. A few reports suggest momentum-indirect dipolar excitons originating from valence bands at the K point and conduction bands at the Q point [40,58,59], despite the strong evidence of momentum-direct dipolar excitons with conduction and valence bands located at the K point [20,47,48,49,50,51,60,61] with a distinct valley Zeeman signature of ∼−16. Interestingly, it has recently been shown that these different dipolar exciton species exhibit different electric dipole moments [45], thus providing a possible alternative with which to properly distinguish them under applied electric fields. In order to fully exploit the functionalities of dipolar excitons in van der Waals heterostructures, it is particularly important to understand their spin-valley physics, and how they depend on external parameters, such as applied electric fields [36,62] and fluctuations of the interlayer distance [63].

In this manuscript, we employ first-principles calculations, to study the spin-valley physics of MoSe${}_{2}$/WSe${}_{2}$ heterostructures, focusing on high-symmetry stackings with 0° (R-type) and 60° (H-type) twist angle under the influence of external electric fields and variations of the interlayer distance. We perform a systematic analysis of the spin ($S_{z}$) and orbital ($L_{z}$) angular momenta, g-factors, and symmetry properties of the relevant band edges (at K, Q, and Γ points), and reveal that different stackings imprint distinct hybridization signatures on the spin-valley physics of these low-energy bands. We follow the successful framework for computing the valley Zeeman splitting in TMDC materials [64,65,66,67], in which atomistic and “valley” contributions are correctly taken into account via the orbital angular momentum of the Bloch function. We extend our analysis to investigate direct- and indirect-in-momentum dipolar (interlayer) excitons, revealing that different exciton species present distinct electric dipole moments, polarizabilities, and g-factors. In particular, direct dipolar excitons in the K-valley hold a robust valley Zeeman splitting nearly independent of the electric field, whereas momentum-indirect dipolar excitons, with conduction bands at the Q point, show a clear dependence of the valley Zeeman splitting on the positive values of applied electric fields (with variations of ∼4). The dipolar exciton g-factors can also be affected by varying the interlayer distance, which could be experimentally probed under external pressure. Because of the opposite orientation of $S_{z}$ and $L_{z}$ of the individual monolayers, our results reveal that changes in the valley Zeeman, due to electric field or interlayer distance, are more pronounced in H-type systems, such as the g-factor variation from −5 to 3 in the valence bands of the H${}_{h}^{h}$ stacking. These pronounced valley Zeeman signatures should be experimentally accessible in samples with a twist angle close to 60°, which give rise to moiré or atomic reconstructed domains dominated by the high-symmetry H-type stackings.

This manuscript is organized as follows: in Section 2, we discuss the electronic structure and symmetry properties of the studied R- and H-type high-symmetry stackings; in Section 3, we explore the spin-valley physics of the relevant band edges, and how they are affected by an applied electric field and variations of the interlayer distance; in Section 4, we focus on the direct and indirect, in *k*-space, dipolar excitons originated from the low-energy band edges; we discuss their selection rules, based on symmetry analysis and first-principles calculations, their valley Zeeman signatures (effective g-factors), and how the effect of external electric fields and variations of the interlayer distance impact the optical selection rules and valley Zeeman physics; finally, in Section 5, we summarize our findings, and place our study in the broader context of the growing research field of valleytronics in multilayered van der Waals heterostructures.

## 2. General Features of MoSe${}_{2}$/WSe${}_{2}$ High-Symmetry Stackings

We focus on artificially stacked MoSe${}_{2}$/WSe${}_{2}$ heterostructures with relative angles of 0° (R-type) and 60° (H-type). When the individual MoSe${}_{2}$ and WSe${}_{2}$ layers are stacked together, the angle is not precisely 0° or 60°, and the small misalignment can give rise to moiré superlattices or atomic reconstruction [19,20,52,53,54,55,56,57], with length scales of tens of nm. However, either in the moiré or in the atomic reconstruction perspective, there are distinct contributions from well-defined high-symmetry stackings that dominate the observed properties [20,64,68,69,70], such as the dipolar excitons selection rules and g-factors. Furthermore, these high-symmetry stackings also serve as building blocks for the effective modeling of interlayer moiré excitons [68,69,71,72,73,74,75].

Side-views of the considered high-symmetry stackings (R${}_{h}^{h}$, R${}_{h}^{M}$, R${}_{h}^{X}$, H${}_{h}^{h}$, H${}_{h}^{M}$, and H${}_{h}^{X}$) are shown as insets in Figure 1a–f, and we follow the labeling of Refs. [19,64]. To clarify, the stacking label X${}_{b}^{a}$ indicates the twist angle X = R (0°) or X = H (60°) and relative shifts between the bottom (subscript a) and top (superscript b) layers, with a,b = h (hollow), M (metal), and X (chalcogen). For instance, the R${}_{h}^{X}$ stacking indicates a structure with 0°, with the chalcogen atom of the top layer positioned on top of the hollow position of the bottom layer. To construct the heterostructure, we start with the values of the in-plane lattice parameter (*a*) and the thickness (*t*) of the individual monolayers, given by a=3.289Å, t=3.335Å for MoSe${}_{2}$, a=3.282Å, t=3.340Å for WSe${}_{2}$, taken from Ref. [76]. For the MoSe${}_{2}$/WSe${}_{2}$ heterostructures, we consider an average lattice parameter of a=3.2855Å, leading to an average strain of ∼0.1 %. We also correct the monolayer thickness due to strain [77], leading to t=3.3375Å and t=3.3374Å for MoSe${}_{2}$ and WSe${}_{2}$, respectively. A vacuum region of 20Å is considered for all cases. The equilibrium interlayer distances, *d*, are given in Table 1, and are consistent with several reports in the literature [19,37,38,64,78,79,80]. We also present, in Table 1, the effective distance between Mo and W atoms. We perform our first-principles calculations based on the density functional theory (DFT), as implemented in the WIEN2k [81] code, with details given in Appendix A.

The layer-resolved band structures for all the calculated R- and H-type structures are shown in Figure 1a–f. We are particularly interested in the band edges that can host dipolar excitons (K, Q, and Γ points), as depicted in Figure 1g, and how they are affected by external electric fields and changes of the interlayer distance, schematically shown in Figure 1h. The conduction and top valence bands at the K point are highly layer-localized, and maintain the same order, regardless of the stacking (bands $c_{W\pm }$, $c_{M\pm }$, $v_{W}$, and $v_{M}$). On the other hand, the ordering of the lowest valence bands depends on the particular stacking (therefore, we identify them as just $v_{+}$ and $v_{-}$, specifying the majority layer composition when necessary). The energy bands outside of the K point often show a higher interlayer hybridization, as is the case of the low-energy conduction bands at the Q point ($c_{Q\pm }$), and the top valence band at the Γ point ($v_{\Gamma }$). Because of the different twist angle (0° or 60°) between the layers, the relative spin degree of freedom is certainly affected, as indicated by the spin-resolved band structure shown in Figure 2.

In terms of symmetry, the R- and H-type structures considered belong to the $C_{3v}$ (P3m1) point group. In reciprocal space, the symmetry group of the Γ point is also the $C_{3v}$ group, and is reduced to its subgroups for other *k*-points. We find the $C_{3}$ group for the K point, the $C_{1}$ group for the Q point, and the $C_{s}$ group for the M point. It is also relevant to identify the double-group (with SOC) irreducible representations (irreps) for the relevant energy bands shown in Figure 1g: these irreps are useful building blocks with which to study the allowed optical selection rules (direct and phonon-mediated), and possible (intra- and/or inter-valley) scattering mechanisms, as already studied in the monolayer cases [50,82,83,84,85]. Furthermore, irreps are crucial to building effective models within the **k.p** framework [76,86,87,88,89,90], in which additional symmetry-breaking terms can be easily incorporated. Specifically for the K point, the irreps of the energy bands are given in Table 2. As the $K_{4}$ and $K_{5}$ irreps are complex, the irreps at −K can be found by taking the complex conjugate $K_{4}\to K_{5}=K_{4}^{*}$, $K_{5}\to K_{4}=K_{5}^{*}$. The top valence band at the Γ point (2-fold degenerate) belongs to the (real) irrep $\Gamma_{4}$ of the $C_{3v}$ group, and the lower conduction bands at the Q point both belong to the (real) irrep $Q_{2}$ of the $C_{1}$ group. To clarify our notation, we identify the irreps by their reciprocal space point, i.e., K${}_{i}$ irreps belong to the K point, $\Gamma_{i}$ irreps for the Γ point, and Q${}_{i}$ irreps for the Q point. We also emphasize that the irreps are obtained using the full wave function calculated within DFT, as implemented in WIEN2k [81]. All the irreps and symmetry groups discussed here follow the character tables of Ref. [91].

## 3. Spin-Valley Physics at the Band Edges

### 3.1. Pristine Heterostructures

In this section, we explore the spin and orbital degrees of freedom for the relevant band edges (indicated in Figure 1g) at zero electric field and at the equilibrium interlayer distance. We follow recent first-principles developments, to compute the orbital angular momenta in monolayer TMDCs [64,65,66,67], which has also been successfully applied to investigate a variety of more complex TMDC-based systems and van der Waals heterostructures [27,53,57,80,85,92,93,94,95,96,97,98,99]. We do not aim to provide a detailed description of the methodology here, but briefly summarize the main points. An external magnetic field in the out-of-plane (*z*) direction (same orientation as the electric field in Figure 1h) modifies the energy levels as
(1)$$\begin{array}{cc} E_{\mathrm{ZS}}\left(n,\vec{k}\right) & =g_{z}\left(n,\vec{k}\right)\mu_{B}B\\ \; & = [S_{z}\left(n,\vec{k}\right)+L_{z}\left(n,\vec{k}\right)]\mu_{B}B\;, \end{array}$$
in which the g-factor, $g_{z}\left(n,\vec{k}\right)$, of the Bloch band (labeled by *n* and $\vec{k}$) is written in terms of the spin and orbital angular momenta, given by
(2)$$\begin{array}{cc} S_{z}\left(n,\vec{k}\right) & =\langle n,\vec{k}\;\left|\sigma_{z}\right|n,\vec{k}\rangle \\ L_{z}\left(n,\vec{k}\right) & =\frac{1}{im_{0}}\sum_{m\neq n}\;\frac{P_{x}^{n,m,\vec{k}}P_{y}^{m,n,\vec{k}}\;-\;P_{y}^{n,m,\vec{k}}P_{x}^{m,n,\vec{k}}}{E\left(n,\vec{k}\right)-E\left(m,\vec{k}\right)}\;, \end{array}$$
in which $\sigma_{z}$ is the Pauli matrix for the spin components along *z*, $P_{\alpha }^{n,m,\vec{k}}=\langle n,\vec{k}\;\left|p_{\alpha }\right|m,\vec{k}\rangle$ (α=x,y,z), $\vec{p}$ is the momentum operator, $E(n,\vec{k})$ is the energy of the Bloch band, and $m_{0}$ is the free electron mass. In the expression of the spin angular momentum, we use $g_{0}\approx 2$ for the electron spin g-factor. The summation-over-bands that characterizes $L_{z}$ requires a sufficiently large number of bands (∼500) to achieve convergence, which is properly taken into account in our DFT calculations (see details in Appendix A). These MoSe${}_{2}$/WSe${}_{2}$ heterostructures were investigated in Refs. [53,57,64,67,80] but here we extend our analysis to incorporate more bands, and later investigate the dependence of electric field and interlayer distance. We emphasize here that the electronic and spin properties calculated with the DFT also provide a reliable benchmark for further studies, as WIEN2k [81] is one of the most accurate DFT codes available [100], and is particularly suitable for studying the spin-physics of 2D materials and van der Waals heterostructures [23,24,27,95,101,102,103,104,105].

Our calculated values of $S_{z}$ and $L_{z}$ for the monolayers are collected in Table 3, and the results for the R- and H-type stackings are shown in Table 4. For the monolayer cases, we use the same in-plane lattice parameters and thicknesses as used in the heterostructure, so that we can directly compare the monolayer values of $S_{z}$ and $L_{z}$ to the heterostructure. The amount of strain of ∼0.1% for the in-plane lattice parameter has a very small influence on the spin-valley physics of individual monolayer TMDCs [77,95], with contributions in the order of $10^{-2}$ from the unstrained values; therefore, any changes of $L_{z}>10^{-2}$ are clear signatures of interlayer hybridization. For the heterostructure bands at the K point, $c_{W(\mathrm{Mo})\pm }$, $v_{W(\mathrm{Mo})}$, and $v_{\pm }$, we observe the typical sign changes from R to H stackings, because of the opposite K/K ($0^{\circ }$ twist) and K/−K ($60^{\circ }$ twist) alignment [20,64]. For the conduction bands, we observe deviations of $L_{z}$ in the order of ≤0.05, when compared to the monolayer case, consistent with a higher localization of conduction bands [27]. We therefore would expect larger changes arising in the valence bands, which is indeed the case, particularly for the $v_{\pm }$ bands, and for the $v_{W}$ and $v_{\mathrm{Mo}}$ bands of the H${}_{h}^{h}$ stacking. Furthermore, the effects of band mixing in the R stacking are less pronounced, because at zero field and at the equilibrium distance the values of $L_{z}$ have the same sign and similar magnitude, whereas in the H stacking, $L_{z}$ has the opposite sign, and any admixture of the energy bands can significantly alter $L_{z}$. For the highly delocalized bands at Γ and Q points, we find values of $S_{z}$ and $L_{z}$ that are in between the values of the individual MoSe${}_{2}$ and WSe${}_{2}$ monolayers, but still with $\left|S_{z}\right|>\left|L_{z}\right|$, thus making spin the dominant degree of freedom, as in the monolayer case [95]. The g-factor values, $g_{z}$, can be found by adding $S_{z}$ and $L_{z}$ (we also present $g_{z}$ graphically in the Section 3.2 and Section 3.3, as a function of the electric field and interlayer distance).

### 3.2. Electric Field Dependence

We now investigate the effect of external electric fields on the band edges of the R- and H-type stackings. In Figure 3, we show the energy dependence of the relevant energy bands (see Figure 1g) as a function of the electric field, with the color code of the K bands representing the layer-decomposition in the top row and the spin degree of freedom in the bottom row. Due to the particular symmetry of the energy bands (orbital and spin characters), the electric field can induce crossings or anti-crossings. The anti-crossings, visible in the top row of Figure 3 when the color code deviates from red (WSe${}_{2}$ layer) or blue (MoSe${}_{2}$ layer), indicate regions with strong interlayer hybridization; furthermore, these anti-crossings can either mix or preserve the spin degrees of freedom, depending on the stacking and bands involved. For instance, for the H${}_{h}^{h}$ stacking, there is an anti-crossing between $v_{M}$ and $v_{-}$ (mainly localized at the WSe${}_{2}$ layer) that preserves spin. On the other hand, the anti-crossing of the R${}_{h}^{M}$ stacking between $v_{M}$ and $v_{-}$ (mainly localized at the WSe${}_{2}$ layer) also acts on the spin degree of freedom. We point out that the conduction bands at the Q point, shown by short dashed lines, do not present any crossing when the energy levels get closer as the values of the electric field increase, as shown in the insets of Figure 3g–l. The crossing of conduction bands takes place when the electric field values are smaller than ∼−0.75 V/nm.

The electric field dependence of the quantities $S_{z}$, $L_{z}$, and $g_{z}$, which encode the essence of spin-valley physics, is summarized in Figure 4, Figure 5 and Figure 6, respectively. The most important feature revealed by our calculations is that $S_{z}$, $L_{z}$, and $g_{z}$ are strongly mixed around the anti-crossing regions shown in Figure 3. In other words, spin and orbital degrees of freedom provide a clear quantitative signature for the strength of the interlayer coupling in these van der Waals heterostructures. Let us discuss in detail the case of H${}_{h}^{h}$ stacking, which shows the most prominent anti-crossing region for the valence bands $v_{M}$ and $v_{-}$. The orbital and spin analysis in Figure 3 reveal that $v_{M}$ and $v_{-}$ mix on the orbital level, while the spin remains unchanged. Indeed, by inspecting Figure 4j, we confirm that $S_{z}$ is unchanged for the interval of the applied electric field. On the other hand, Figure 5j shows that the values of $L_{z}$ are drastically modified as the electric field swipes through the anti-crossing region, with the strongest mixing happening at ∼1.5 V/nm. In fact, already at zero electric field (as shown in Table 4), we clearly observe the hybridization effect taking place in the values of $L_{z}$, suggesting that small values of applied electric field are enough to tune this hybridization even more. We point out that there are anti-crossing signatures also taking place in the conduction band; however, it seems that this feature has not been observed experimentally, due to the range of electric field values applied [42,43,45]. Additionally, the electric field introduces a strong change of the spin character in the conduction bands at the Q point, due to their anti-crossing (insets of Figure 3g–l), an interesting feature that can be investigated in low-energy interlayer excitons [45], which we discuss in further detail in Section 4.3. We note that the field-driven Rashba SOC is naturally incorporated within our first-principles calculations; however, the electric field is just too weak to effectively remove spins from their out-of-plane direction.

Our detailed first-principles calculations reveal that the H${}_{h}^{h}$ stacking is the most suitable structure with which to investigate these interlayer hybridization signatures in the coupled spin-valley degrees of freedom. Surprisingly, this is the most stable structure for $60^{\circ }$ twist angles, and the most relevant stacking for small twist angles, either having in mind a moiré or an atomically reconstructed sample, which facilitates the experimental accessibility to these effects. Furthermore, the opposite orientation of spin and orbital angular momenta from the opposite K-valleys in the individual monolayers favors g-factor variations from negative to positive values: for example, from −3 to 4 in the conduction band (Figure 6d), from −5 to 3 in the valence band (Figure 6j), and from −1 to 1 for conduction bands at the Q point (Figure 6p).

### 3.3. Interlayer Distance Variation

In this section, we investigate the consequences of varying the equilibrium interlayer distance. This can be particularly relevant in the context of fluctuations introduced by the mechanical exfoliation and stamping procedures in TMDC heterostructures [47,56,63], which create regions with good and bad *interlayer contact*. Additionally, changes in the interlayer distance can provide insights into the effects of external out-of-plane pressure, which can be currently investigated in available experimental setups, such as the recently reported studies on graphene/WSe${}_{2}$ [106], multilayered TMDCs [107], and WSe${}_{2}$/Mo${}_{0.5}$W${}_{0.5}$Se${}_{2}$ heterostructures [108]. On a more abstract and theoretical level, artificially varying the interlayer distance provides an additional knob with which to modify the interlayer hybridization [109] and analyze its impact on the spin and orbital angular momenta.

The influence of the interlayer distance variation on the energy levels, spin angular momenta, orbital angular momenta, and g-factors is presented in Figure 7, Figure 8, Figure 9, and Figure 10, respectively. We restrict ourselves to fluctuations of ±20% of the equilibrium interlayer distances given in Table 1. Overall, the same reasoning we discussed for the electric field effect applies to the interlayer distance variations. Nonetheless, we highlight a few points, to strengthen the discussions and understanding presented in the previous Section 3.2. There is a strong orbital mixing region visible in the lower valence bands with same spins in the R${}_{h}^{h}$ stacking case (Figure 7a,g), but with a negligible signature in the $L_{z}$ (Figure 9g), as the monolayer values of $L_{z}$ are all positive and close to each other. On the other hand, these hybridization effects are much stronger in the valence band of the H${}_{h}^{X}$ stacking case (Figure 7f,l), with strong signatures in the values of $S_{z}$ and $L_{z}$ (Figure 8l and Figure 9l). We also note that in the H${}_{h}^{h}$ stacking, the most favorable for the 60° case, the $g_{z}$ values of the valence band (particularly $v_{W}$) can be effectively altered by reducing the interlayer distance (also seen in the H${}_{h}^{M}$ stacking).

## 4. Low-Energy Dipolar Excitons

In this section, we investigate the fundamental ingredients that characterize low-energy dipolar excitons: the electric dipole moment; polarizability; dipole matrix elements; and effective g-factors, based on the robust DFT calculations we presented in Section 2 and Section 3. External factors that can alter the energetic landscape of such dipolar excitons, and consequently their photoluminescence or reflectance spectra, are beyond the scope of the present manuscript. Regarding the spin-valley physics and g-factors, previous studies focusing on intralayer excitons in monolayer TMDCs show that DFT calculations provide remarkable agreement with experimental data for pristine [110,111] and strained [85,94,95] cases. These intralayer excitons have larger conduction-to-valence band dipole matrix elements, larger electron-hole exchange contributions, and weaker dielectric screening, when compared to interlayer excitons. Therefore, if DFT is indeed capturing the relevant spin-valley properties of intralayer excitons, it will be sufficient to provide a reliable description of the spin-valley properties in dipolar excitons. In fact, DFT studies on interlayer exciton g-factors [64,66,67] are indeed capable of providing reliable comparison with experiments.

### 4.1. Symmetry-Based Selection Rules

Starting from the symmetry perspective, we can use the irreps computed in Section 2 to characterize the optical selection rules of the possible interlayer excitons. Let us first discuss the selection rules of direct excitons at the K points, in order to compare with and validate our DFT calculations in the next Section, Section 4.3. Direct optical transitions are characterized by the matrix element mediated by the momentum operator:(3)$$\langle i\left|\vec{p}\cdot \hat{\epsilon }\right|f\rangle ∼I_{i}^{*}⊗I_{p}⊗I_{f}\;,$$
in which $I_{i(f)}$ is the irrep of the initial (final) state, and $I_{p}$ is the irrep of the momentum operator for a specific polarization ($\hat{\epsilon }=\left\{\sigma_{+},\sigma_{-},z\right\}$). The components of the momentum operator for circularly and linearly polarized light in the $C_{3}$ group transform as
(4)$$K_{1}∼z,\;K_{2}∼\sigma_{-},\;K_{3}∼\sigma_{+}\;.$$

Combining Equation (Equation 4) with the irreps of the valence and conduction bands given in Table 2, we can compute the optical selection rules for K-K dipolar excitons. The results are shown in Table 5, and agree with previous symmetry-based and DFT calculations [64,69,73]. It is also possible to verify that the *intralayer* optical selection rules are nicely preserved. For instance, transitions from $v_{W}\left(K\right)$ to $c_{W+}\left(K\right)$ provide $\sigma_{+}$, whereas transitions from $v_{M}\left(K\right)$ to $c_{M-}\left(K\right)$ provide $\sigma_{+}$ for R-type stackings and $\sigma_{-}$ for H-type stackings. An important point to discuss here is related to the so-called *spin-dark* optical transitions (sometimes also refereed to as *spin-flip* or *spin-singlet* transitions): for instance, the $v_{W}\left(K\right)\to c_{M-}\left(K\right)$ transition in the H${}_{h}^{h}$ stacking. The orientation of $S_{z}$, given in Table 3 and Table 4, is not indicative of allowed or forbidden optical transitions: in fact, $S_{z}\neq 1$ if we increase the number digits after the decimal point in Table 3 and Table 4. This is already evident in the monolayer case, due to the existence of the dark excitons [112,113,114]. The main point here is that SOC always introduces a mixing of spins (even in graphene [104]), and that the momentum matrix element does not flip spin (more discussions about spin-mixing and optical transitions for *spin-dark* states in monolayer TMDCs can be found in Ref. [95]). Furthermore, some of the interlayer optical transitions presented in Table 5 have the same structure as the dark/gray excitons in monolayers [112,113,114], i.e., one transition with *z* polarization and the other with $\sigma_{\pm }$. This feature actually explains the energy splitting of ∼0.1 meV experimentally observed at zero magnetic field by Li and coauthors [51] in atomically reconstructed MoSe${}_{2}$/WSe${}_{2}$ heterobilayers, akin to the dark-gray exciton splitting of ∼0.5 meV in pristine W-based monolayers [112,113].

For optical transitions involving momentum-indirect dipolar excitons, i.e., transitions involving valence and conduction bands at different $\vec{k}$ points, a phonon is required, to ensure momentum conservation. The optical transition matrix element mediated by phonons is written as [115]
(5)$$\sum_{l}\langle i\left|H_{e-\mathrm{ph}}\right|l\rangle {\left(E_{f}-E_{l}\right)}^{-1}\langle l\left|\vec{p}\cdot \hat{\epsilon }\right|f\rangle ∼I_{i}^{*}⊗I_{\mathrm{ph}}⊗I_{p}⊗I_{f}\;,$$
in which $H_{e-\mathrm{ph}}$ takes into account the electron-phonon coupling, and transforms as the irrep $I_{\mathrm{ph}}$. It is beyond the scope of this manuscript to provide deeper insights into the strength of the electron-phonon coupling: rather, we discuss qualitatively the possible symmetry-allowed mechanisms involving the momentum-indirect dipolar excitons.

For indirect excitons with electrons and holes in opposite K valleys (−K and K, sharing the same $C_{3}$ group), the direct product $I_{i}^{*}⊗I_{f}$ provides the output shown in Table 6. Note that the output is always restricted to the irreps $K_{1}$, $K_{2}$, and $K_{3}$. Phonons in the $C_{3}$ group would span all the irreps of the simple group, namely $K_{1}$, $K_{2}$, and $K_{3}$. Combining the results given in Table 6, and performing a direct product with all possible phonon modes, we obtain
(6)$$\begin{array}{cc} K_{1}⊗\left\{K_{1},K_{2},K_{3}\right\} & =\left\{K_{1},K_{2},K_{3}\right\}\\ K_{2}⊗\left\{K_{1},K_{2},K_{3}\right\} & =\left\{K_{2},K_{3},K_{1}\right\}\\ K_{3}⊗\left\{K_{1},K_{2},K_{3}\right\} & =\left\{K_{3},K_{1},K_{2}\right\} \end{array}$$
which always provides $K_{1}$, $K_{2}$, and $K_{3}$. These irreps also encode the light polarization, as given in Equation (Equation 4). What our results suggest is that any type of phonon with momentum K can, in principle, mediate the optical recombination of dipolar excitons with electrons and holes residing in opposite K valleys. Perhaps some of these processes are not favored, due to off-resonant conditions and weak electron-phonon coupling, but could be effectively triggered when phonon energies are brought into resonance, such as was the case of the 24T anomaly [47,50,61]. Certainly, these phonon processes are intimately dependent on the strength of the electron-phonon couplings and how those couplings are translated for the final excitonic state (based on the particular conduction and valence bands constituents).

To investigate indirect dipolar excitons with holes at the Γ point and electrons at the K valley that do not have the same symmetry group, we must describe both conduction and valence band states on the same footing, using compatibility relations [89]. It is more convenient to go from a higher symmetry group to a lower symmetry group then the other way around. With this in mind, we mapped Γ point states to the K valley $\Gamma_{4}\to K_{4}⊕K_{5}$ and present the direct product in Table 7. The same argument discussed in the case of electrons and holes at opposite valleys applies here, and multiple phonons with momentum K could trigger the optical recombination of such Γ-K excitons.

Finally, we discuss the indirect excitons involving conduction bands at the Q point and holes at either Γ or K points. As the $C_{1}$ symmetry group contains a single irrep in the simple group, all electronic states, phonon modes, and operators are mapped onto this irrep: consequently, multiple phonon mediated processes are allowed. We note that Q–K transitions have been recently observed [45], supporting the idea that some of these symmetry-allowed phonons discussed here are indeed favoring the optical recombination.

### 4.2. Effective g-Factors

Combining the band g-factors studied in Section 3 with the optical selection rules discussed in Section 4.1, we investigate here the effective g-factors of the relevant dipolar excitons. As the momentum-indirect excitons require a phonon to mediate the optical emission, we can only determine unambiguously the sign of the g-factors for direct interlayer excitons, while for the other dipolar states we restrict ourselves to the magnitude of the g-factors. The dipolar exciton g-factors can be written as
(7)$$g\left(X\right)=2\left|g(c)\pm g(v)\right|\;,$$
in which g(c) and g(v) represent the conduction and valence band g-factors, and the ± sign contemplates the two possibilities of time-reversal partners. For example, interlayer excitons with the valence band $v=v_{W}$ and the conduction bands $c=c_{\mathrm{Mo}\pm }$ could be generated with *v* and *c* at the K point (direct in momentum), as indicated in Figure 1g, or with *v* in the K point and *c* in the −K point (indirect in momentum).

The calculated g-factor values for the studied dipolar excitons are shown in Table 8. The largest absolute values stem from the KK dipolar excitons (*v* at K and *c* at K in H-type, *v* at −K and *c* at K in R-type), as they involve the energy bands with the largest g-factors. Some dipolar excitons show very similar g-factors, such as the values related to $v_{W}\left(-K\right)\to c_{M+}\left(K\right)$ and $v_{W}\left(-K\right)\to c_{M+}\left(K\right)$ excitons in H-type stackings. We also observe g-factors with values <1, and even ∼0 [$v_{\Gamma_{1}}\to c_{Q-}\left(Q\right)$], mainly originating from the valence bands at the Γ point. These different g-factors originate from different stackings, and distinct dipolar exciton species can be observed in the crossover region between deep and shallow moiré potentials, as a function of the twist angle or temperature; however, no systematic experimental study has been realized to explore such conditions. We emphasize that our calculated values are in excellent agreement with previously reported theoretical studies [57,64,67,80], and lie within the available experimental values, which range from −15 to −17 in H-type stackings [20,47,48,49,50,51,53,60,61], and from 5 to 8 in R-type stackings [20,42,48,49,50,51,116,117].

### 4.3. Electric Field Dependence

We now turn to the behavior of the different dipolar excitons as a function of the electric field. Motivated by the experimentally accessible regime [42,43,44,45,46], in which there is no indication of any crossings in the top valence band or lower conduction bands, we restrict ourselves to electric field values ≳−0.75 V/nm, ensuring that the conduction band of MoSe${}_{2}$ is always below the conduction band of WSe${}_{2}$. For positive values of electric field, there are no visible band crossings, so we can extend our analysis towards electric field values of ∼2 V/nm. In fact, some experimental studies that investigate dipolar excitons under electric field also show an asymmetric range [42,43,45], supporting the electric field values we consider here.

The energy dependence of the dipolar excitons under external electric fields can be directly inferred from the dependence of the energy bands, because the effective mass is barely affected by the electric field (changes on the order of $10^{-3}$–$10^{-2}$), and the dielectric environment remains unchanged; therefore, the energy variation of the dipolar excitons with respect to the electric field can be extracted directly from the DFT calculations [63], and modeled by the following dependence [118,119]:(8)$$\Delta E\left(F_{z}\right)=-\mu F_{z}-\frac{\alpha }{2}F_{z}^{2}\;,$$
in which μ is the electric dipole moment (in meV nm/V), α is the polarizability (in meV nm${}^{2}$/V${}^{2}$), and $F_{z}$ is the electric field in V/nm.

Our calculated values of the electric dipole moment and polarizabilities, fitted in the range of −0.7 to 0.7 V/nm, are presented in Figure 11. Overall, our results reveal that the values of μ (α) decrease (increase) as we go from momentum-direct to momentum-indirect dipolar excitons (KK → KQ →ΓK →ΓQ), reflecting the degree of layer delocalization of the bands involved (Γ> Q > K, as shown in Figure 1a–f). These delocalization features have already been observed in the magnitude of interlayer excitons of WSe${}_{2}$ homobilayer structures [120,121], and there is one report on MoSe${}_{2}$/WSe${}_{2}$ heterobilayers [45]. Our results also show that the spin-split conduction bands $c_{M\pm }$ have negligible effect on μ and α of KK and ΓK excitons, whereas the conduction bands $c_{Q\pm }$ provide different signatures, as they have different delocalization over the two layers. Particularly for the KK dipolar excitons, we note that the calculated values of α are quite small (0.2–2 meV nm${}^{2}$/V${}^{2}$), and that the two possible excitons stemming from $c_{M\pm }$ are nearly indistinguishable under the electric field dependence; however, they would have different emission energies (due to the SOC splitting of $c_{M\pm }$) and distinct g-factor signatures. We note that our calculated values of μ for KK and KQ dipolar excitons—and particularly that $\mu_{\mathrm{KK}}>\mu_{\mathrm{KQ}}$—are consistent with recent GW + Bethe-Salpeter calculations and experiments in MoSe${}_{2}$/WSe${}_{2}$ heterobilayers [45].

To validate the symmetry-based selection rules of the direct interlayer excitons discussed in Section 4.1, we present the dipole matrix elements computed within DFT in Figure 12. All the non-zero values match exactly the symmetry-based arguments discussed in Section 4.4. Generally, the dipole matrix elements decrease for increasing values of the electric field, with the exception of the $p_{c_{M+},v_{W}}$, which slightly increases for increasing electric field, reflecting the impact of the atomic registry on the wave functions of van der Waals heterostacks. Furthermore, the *spin-dark* optical transitions discussed in terms of symmetry in Section 4.1 are also present here. The momentum operator that enters the dipole matrix element (Equation (Equation 4)) does not flip spin, and therefore this transition is optically bright only if the 2-component spinors are mixed, which is indeed the case in our first-principles calculations with SOC. Our calculated values of the dipole matrix elements, along with their respective polarization, can be readily used to generate massive Dirac models for the dipolar excitons via the first-order perturbation within the **k.p** formalism, allowing the investigation of spin-dependent physics beyond the parabolic approximation [73,75].

Based on the calculated values of the energy dependence and dipole matrix elements, with respect to the electric field, we can evaluate the radiative lifetime for direct dipolar excitons, given by [122,123]:(9)$$\tau_{\mathrm{rad}}=\frac{\sqrt{\varkappa_{b}}c\hbar^{2}}{2\pi e^{2}}\frac{E_{0}}{{\left|p_{c,v}\right|}^{2}{\left|\phi (0)\right|}^{2}}\;,$$
with $\varkappa_{b}$ being the high-frequency dielectric constant of the environment, *c* the speed of light, *ℏ* the reduced Plank constant, *e* the electron charge, $E_{0}$ the exciton energy, and ${\left|\phi (0)\right|}^{2}$ the exciton envelope function evaluated at the origin. The relevant quantities that depend on the electric field, $F_{z}$, are the exciton energy, $E_{0}\to E_{0}+\Delta E\left(F_{z}\right)$ (see Equation (Equation 1)), and the dipole matrix element, $p_{c,v}\to p_{c,v}\left(F_{z}\right)$ (see Figure 12). Typical interlayer excitons exhibit lifetimes reaching almost 100 ns at very low temperatures, and a few ns at room temperature [39,40,41], with roughly one order of magnitude fluctuation when comparing different samples. To understand the isolated impact of the electric field on the radiative lifetime of the direct dipolar excitons, we consider the quantity
(10)$$\frac{\tau_{\mathrm{rad}}\left(F_{z}\right)}{\tau_{\mathrm{rad}}\left(0\right)}=\frac{E_{0}+\Delta E\left(F_{z}\right)}{{\left|p_{c,v}\left(F_{z}\right)\right|}^{2}}\frac{{\left|p_{c,v}\left(0\right)\right|}^{2}}{E_{0}}\;,$$
and display its calculated values in Figure 13 as a function of the electric field. The calculated ratio $\tau_{\mathrm{rad}}\left(F_{z}\right)/\tau_{\mathrm{rad}}\left(0\right)$ varies in general in a range from 0.5 to 4.5, except in the case of H${}_{h}^{X}$, which shows a drastic increase in the radiative lifetime, because the dipole matrix element decreases to nearly zero in Figure 12f. Particularly for the high-symmetry stacking H${}_{h}^{h}$, which is the most favorable one for ∼60° samples, both direct dipolar excitons (singlet and triplet) exhibit an increase in the radiative lifetime as the electric field increases, which can be accessed experimentally in current devices.

Let us now discuss electric field effects on the valley Zeeman physics of the dipolar excitons (schematically shown in Figure 1g). The effective g-factors calculated via Equation (Equation 7) as a function of the electric field are shown in Figure 14 for all the considered R- and H-type stackings. Our results reveal that these dipolar excitons have a very robust valley Zeeman, barely changing under external electric field. The only exceptions are the dipolar excitons involving Q bands in the H-type stackings. Because of the g-factor crossing between the $c_{Q+}$ and $c_{Q-}$ bands (see Figure 6p,q,r), stemming from the spin mixing effects (see Figure 4p,q,r and Figure 5p,q,r), the valley Zeeman of these indirect dipolar excitons is affected, and exhibits pronounced changes for increasing electric fields (see Figure 14d–f,j–l,p–r,v,x,y). Therefore, without significant band hybridization, the dipolar excitons retain their valley Zeeman signatures over a large range of applied electric field values, which can be particularly relevant for electrically-driven opto-spintronics applications [17]. In fact, our results reveal very fundamental and general features about the valley Zeeman mixing of excitonic states (including dipolar excitons) in 2D materials and van der Waals heterostructures—in this case, stemming from the band hybridization between different layers. Clear signatures of the valley Zeeman mixing on the excitonic states are still rather limited, but have been already observed experimentally in a few systems: strained WS${}_{2}$ monolayers [85]; defect-mediated excitons in MoS${}_{2}$ monolayers [97,98]; WS${}_{2}$/graphene heterostructures [27]; and electrically-controlled trilayer MoSe${}_{2}$ structures [124].

### 4.4. Interlayer Distance Variation

In this section, we focus on the effect of the interlayer distance variation. Our calculated values of the dipole matrix elements and the g-factors are presented in Figure 15 and Figure 16, respectively, and follow the same convention as discussed in the case of the electric field (Figure 12 and Figure 14). In particular, we find a strong decrease of the dipole matrix elements as the interlayer distance increases, consequently making the dipolar excitons less optically active. These findings provide support to the idea that weakly coupled heterobilayers (with an interlayer separation larger than the equilibrium distance)—with, perhaps, additional electric field displacements due to asymmetric bottom and top environments—may exhibit highly quenched optical emission of dipolar excitons [56,63]; furthermore, as the dipole matrix elements behave in a fashion similar to the electric field, we expect similar behavior for the quantity $\tau_{\mathrm{rad}}\left(F_{z}\right)/\tau_{\mathrm{rad}}\left(0\right)$. Another interesting aspect, visible in the H${}_{h}^{h}$ stacking, is the decrease of the valley Zeeman originating from the top valence band at the K point, $v_{W}$, as the interlayer distance decreases (Figure 16d,j): we believe that this feature could be experimentally probed, using external pressure [106,107], which would effectively decrease the interlayer distance. For the H${}_{h}^{M}$ stacking, we also observe the signatures of Q-band mixing and reversal of the g-factor values as the interlayer distance increases.

## 5. Concluding Remarks

In summary, we have performed detailed first-principles calculations on the spin-valley physics of MoSe${}_{2}$/WSe${}_{2}$ heterobilayers under the effect of out-of-plane electric fields and as a function of the interlayer separation. Specifically, we have investigated the spin and orbital degrees of freedom, g-factors, and symmetry properties of the relevant band edges (located at the K, Q and Γ points) of high-symmetry stackings with 0° (R-type) and 60° (H-type) twist angles. These R- and H-type stackings are the fundamental building blocks present in moiré and atomically reconstructed supercells, experimentally observed in samples with small twist angles. Our calculations reveal distinct hybridization signatures of the spin and orbital degrees of freedom as a function of the electric field and interlayer separation, depending on the energy band and on the stacking configuration. With this crucial knowledge of the spin-valley physics of the band edges, we expanded our analysis to the low-energy dipolar (interlayer) excitons, either direct or indirect in *k*-space. We found that these different dipolar exciton species present distinct electric dipole moments, polarizabilities, and valley Zeeman signatures (g-factors), due to the particular mixing of the spin and orbital angular momenta from their constituent energy bands. We also performed a symmetry analysis to discuss the phonon modes that could mediate the optical recombination of momentum-indirect dipolar excitons. Regarding the valley Zeeman signatures of the interlayer excitonic states, our calculations revealed that direct dipolar excitons at the K-valley carry a robust valley Zeeman effect nearly independent of the electric field. On the other hand, the momentum-indirect dipolar excitons in the H-type stackings, with valence band at the K valleys and conduction bands at the Q valleys, showed a pronounced dependence of the valley Zeeman for positive values on applied electric fields. This peculiar g-factor dependence was unambiguously related to the spin-mixing of conduction bands at the Q point, which was absent in the R-type stackings. The valley Zeeman physics of the investigated dipolar excitons could also be affected by varying the interlayer distance, which could be experimentally probed under external pressure. As in the case of external electric fields, the interlayer variation effects in the valley Zeeman signatures were more pronounced in H-type stackings, and were more likely to be observed in samples with twist angle close to 60°.

Although *intralayer* excitons were not within the scope of the present manuscript, these quasiparticles are still optically active and easily recognized in the photoluminescence or reflectivity spectra of MoSe${}_{2}$/WSe${}_{2}$ heterobilayers [39,47,99]. In particular, our g-factor calculations for the band edges (Figure 6) suggest that *intralayer* excitons could also provide valuable information on the interlayer hybridization via the valley Zeeman as a function of the electric field. Gated devices based on MoSe${}_{2}$/WSe${}_{2}$ heterostructures are within experimental reach [18,42,46], and could be used to test our hypothesis. We note that, in order to properly evaluate the *intralayer* exciton physics, it would be desirable to go beyond DFT calculations, and to employ the GW + Bethe–Salpeter formalism [45,65,97,125], which is beyond the scope of this work. Nonetheless, as these *intralayer* excitons are also quite localized in *k*-space [10,14,27], the spin and orbital angular momenta of the Bloch bands should account for the majority of the hybridization effects.

We emphasize that the theoretical approach employed here in our study is robust, and provides quantitative microscopic insights into the spin-valley physics of these van der Waals heterobilayers. This framework can certainly be extended to more complex multilayered van der Waals heterostructures that also host dipolar excitons: for instance, large-angle twisted MoSe${}_{2}$/WSe${}_{2}$ [126,127], homobilayers [22,125], trilayer hetero- and homo-structures [124,128,129], and MoSe${}_{2}$/WS${}_{2}$ heterobilayers [99,130,131].

## Figures and Tables

**Figure 1 nanomaterials-13-01187-f001:**
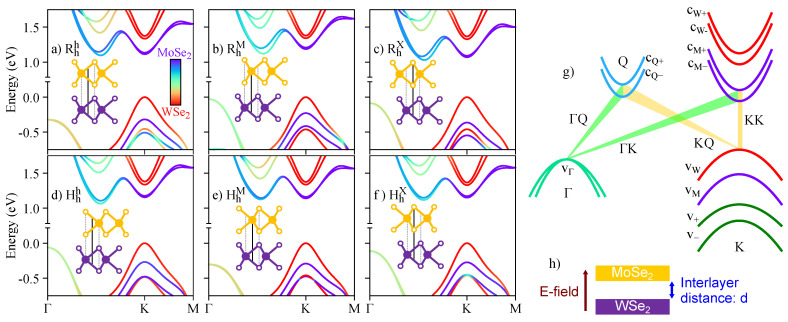
(**a–f**) calculated band structures, with the color-coded layer decomposition of the wave function for (**a**) R${}_{h}^{h}$, (**b**) R${}_{h}^{M}$, (**c**) R${}_{h}^{X}$, (**d**) H${}_{h}^{h}$, (**e**) H${}_{h}^{M}$, and (**f**) H${}_{h}^{X}$ stackings; the insets show the side-view of the crystal structures (solid lines connect the hollow position of the bottom layer to the atomic registry of the top layer); (**g**) relevant low-energy bands and possible interlayer exciton transitions originating from the top valence bands at the Γ (v${}_{\Gamma }$) and K points (v${}_{W}$); the transitions involving the time-reversal partners (−K and −Q points) are not shown, for simplicity; (**h**) schematic representation of the applied external electric field and the interlayer distance.

**Figure 2 nanomaterials-13-01187-f002:**
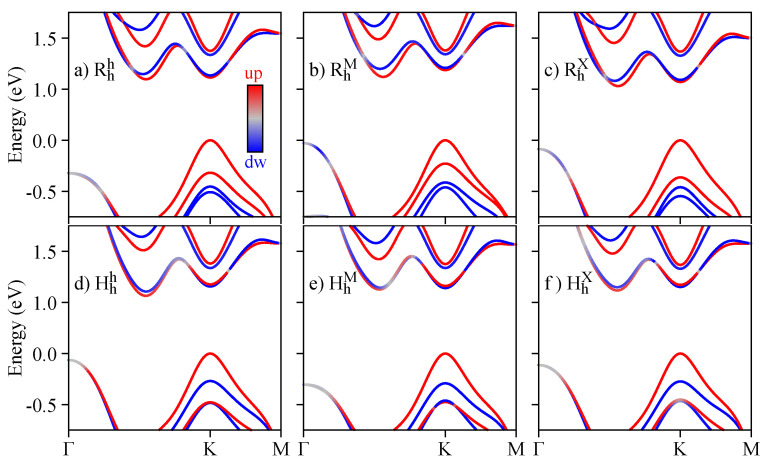
Spin-resolved band structures for the (**a**) R${}_{h}^{h}$, (**b**) R${}_{h}^{M}$, (**c**) R${}_{h}^{X}$, (**d**) H${}_{h}^{h}$, (**e**) H${}_{h}^{M}$, and (**f**) H${}_{h}^{X}$ stackings.

**Figure 3 nanomaterials-13-01187-f003:**
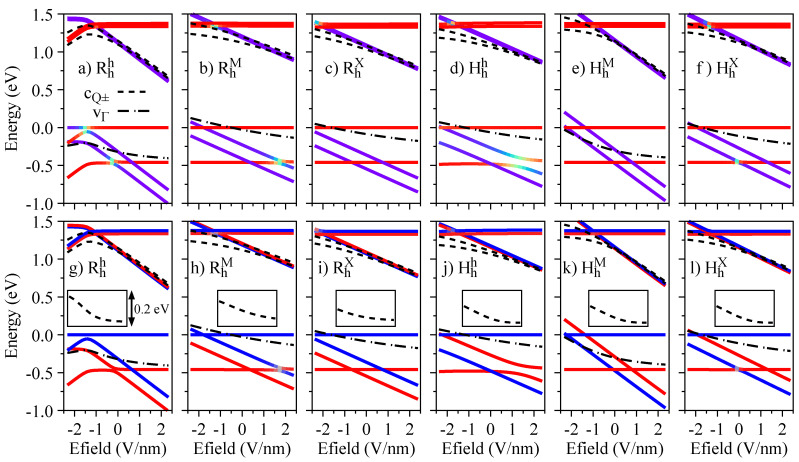
Energy dependence with respect to the electric field of the relevant low-energy bands (see Figure 1g) for all considered stackings: the top row, panels (**a**–**f**), indicates the color-coded layer decomposition of the K point bands (same color code as in Figure 1a–f); the bottom row, panels (**g**–**l**), indicates the color-coded spin decomposition of the K point bands (same color code as in Figure 2a–f); the insets in panels (**g**–**l**) show the energy difference between $c_{Q+}$ and $c_{Q-}$ bands, emphasizing an anti-crossing at larger electric fields.

**Figure 4 nanomaterials-13-01187-f004:**
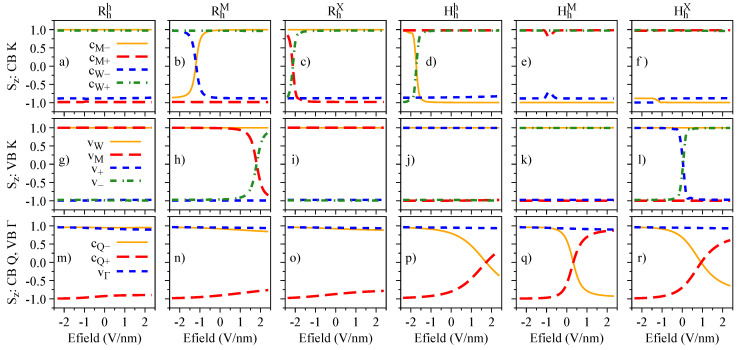
Spin degree of freedom, $S_{z}$, of the low-energy bands as a function of the applied electric field for the studied stackings R${}_{h}^{h}$ (**a**,**g**,**m**), R${}_{h}^{M}$ (**b**,**h**,**n**), R${}_{h}^{X}$ (**c**,**i**,**o**), H${}_{h}^{h}$ (**d**,**j**,**p**), H${}_{h}^{M}$ (**e**,**k**,**q**), and H${}_{h}^{X}$ (**f**,**l**,**r**).

**Figure 5 nanomaterials-13-01187-f005:**
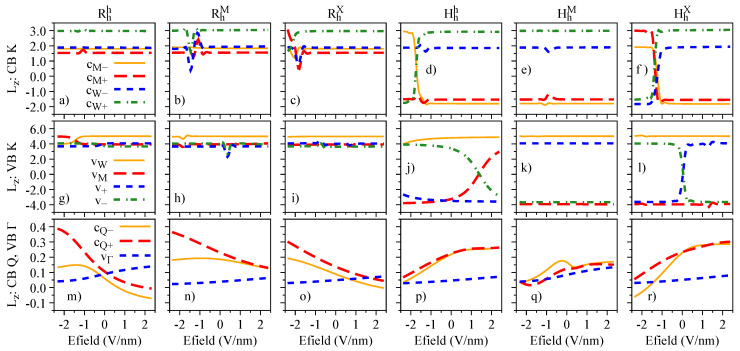
Same as Figure 4 but for the orbital degree freedom, $L_{z}$.

**Figure 6 nanomaterials-13-01187-f006:**
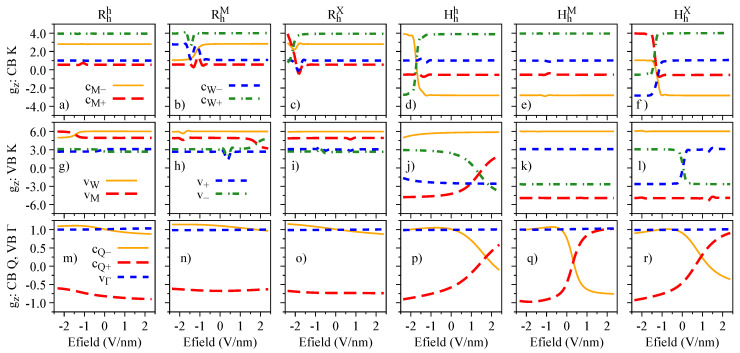
Same as Figure 4 and Figure 5 but for the band g-factor, $g_{z}=L_{z}+S_{z}$.

**Figure 7 nanomaterials-13-01187-f007:**
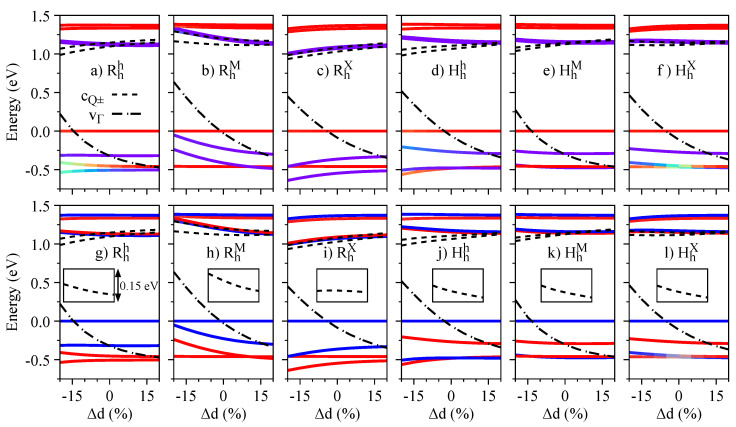
Same as Figure 3 but as a function of the interlayer distance variation.

**Figure 8 nanomaterials-13-01187-f008:**
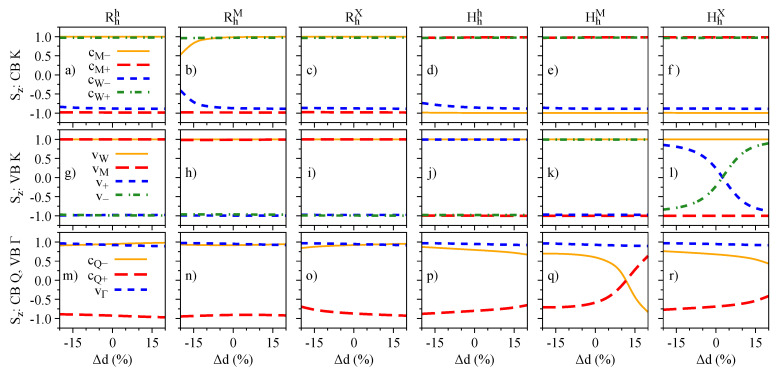
Same as Figure 4 but as a function of the interlayer distance variation.

**Figure 9 nanomaterials-13-01187-f009:**
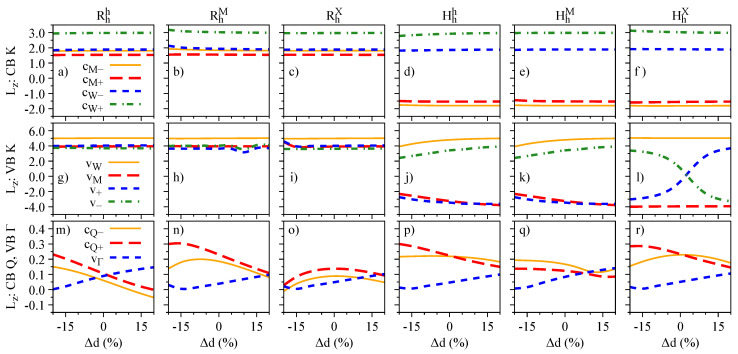
Same as Figure 5 but as a function of the interlayer distance variation.

**Figure 10 nanomaterials-13-01187-f010:**
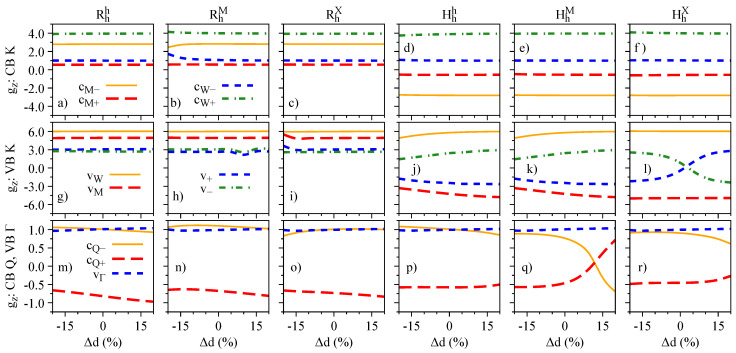
Same as Figure 6 but as a function of the interlayer distance variation.

**Figure 11 nanomaterials-13-01187-f011:**
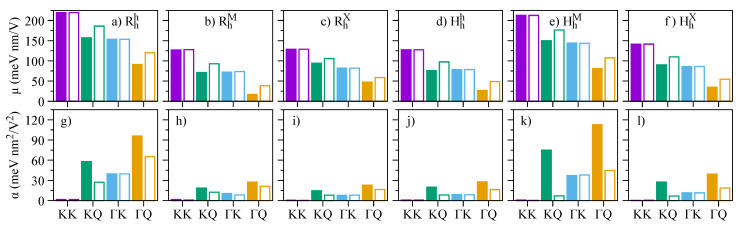
Calculated values of the electric dipole moments for (**a**) R${}_{h}^{h}$, (**b**) R${}_{h}^{M}$, (**c**) R${}_{h}^{X}$, (**d**) H${}_{h}^{h}$, (**e**) H${}_{h}^{M}$, and (**f**) H${}_{h}^{X}$ stackings. Calculated values of the polarizabilities for (**g**) R${}_{h}^{h}$, (**h**) R${}_{h}^{M}$, (**i**) R${}_{h}^{X}$, (**j**) H${}_{h}^{h}$, (**k**) H${}_{h}^{M}$, and (**l**) H${}_{h}^{X}$ stackings. The x-axis indicates the type of dipolar excitons (see Figure 1g. The values originating from $c_{M/Q-}$ ($c_{M/Q+}$) are shown with colored (open) boxes.

**Figure 12 nanomaterials-13-01187-f012:**
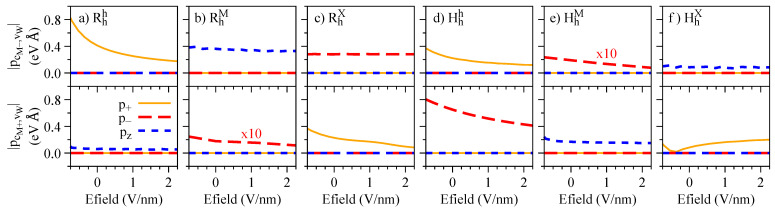
Absolute value of the dipole matrix element for interlayer transitions at the K point as a function of the electric field for (**a**) R${}_{h}^{h}$, (**b**) R${}_{h}^{M}$, (**c**) R${}_{h}^{X}$, (**d**) H${}_{h}^{h}$, (**e**) H${}_{h}^{M}$, and (**f**) H${}_{h}^{X}$ stackings.

**Figure 13 nanomaterials-13-01187-f013:**
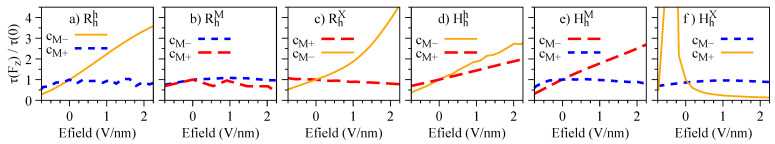
Calculated ratio $\tau_{\mathrm{rad}}\left(F_{z}\right)/\tau_{\mathrm{rad}}\left(0\right)$ as a function of the electric field for (**a**) R${}_{h}^{h}$, (**b**) R${}_{h}^{M}$, (**c**) R${}_{h}^{X}$, (**d**) H${}_{h}^{h}$, (**e**) H${}_{h}^{M}$, and (**f**) H${}_{h}^{X}$ stackings. We use $E_{0}=1.35$ eV for all cases, and the calculated values of μ given in Figure 11. The contribution of α is neglected, as they nearly vanish for KK dipolar excitons.

**Figure 14 nanomaterials-13-01187-f014:**
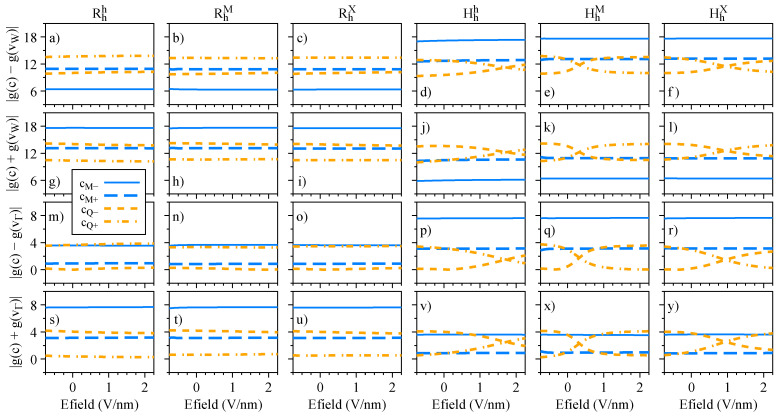
Dipolar exciton g-factors as a function of the electric field for the R- and H-type systems studied for the cases (**a**–**f**) $\left|g\left(c\right)-g\left(v_{W}\right)\right|$, (**g**–**l**) $\left|g\left(c\right)+g\left(v_{W}\right)\right|$, (**m**–**r**) $\left|g\left(c\right)-g\left(v_{\Gamma }\right)\right|$ and (**s**–**v**,**x**,**y**) $\left|g\left(c\right)+g\left(v_{\Gamma }\right)\right|$.

**Figure 15 nanomaterials-13-01187-f015:**
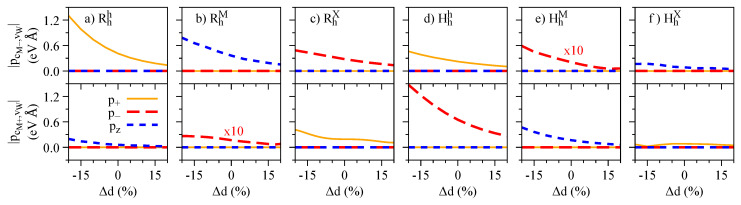
Same as Figure 12 but as a function o the interlayer distance.

**Figure 16 nanomaterials-13-01187-f016:**
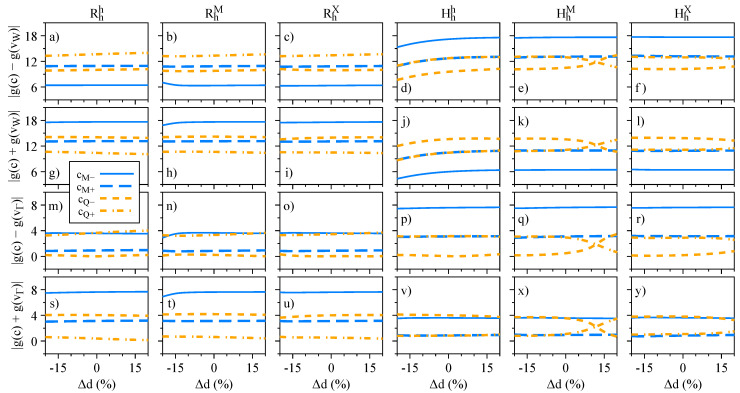
Same as Figure 14 but as a function of the interlayer distance.

**Table 1 nanomaterials-13-01187-t001:** Interlayer distance, d, and effective Mo-W distance for the studied R- and H-type systems.

	R${}_{h}^{h}$	R${}_{h}^{M}$	R${}_{h}^{X}$	H${}_{h}^{h}$	H${}_{h}^{M}$	H${}_{h}^{X}$
d (Å)	3.7237	3.0803	3.0869	3.0923	3.6885	3.1833
Mo-W (Å)	7.0612	6.6922	6.6985	6.7037	7.2775	6.5208

**Table 2 nanomaterials-13-01187-t002:** Irreducible representations at the K point ($C_{3}$ point group) for the relevant energy bands indicated in Figure 1g for all considered stackings.

	R${}_{h}^{h}$	R${}_{h}^{M}$	R${}_{h}^{X}$	H${}_{h}^{h}$	H${}_{h}^{M}$	H${}_{h}^{X}$
$c_{W+}$	$K_{5}$	$K_{6}$	$K_{4}$	$K_{4}$	$K_{6}$	$K_{5}$
$c_{W-}$	$K_{4}$	$K_{5}$	$K_{6}$	$K_{6}$	$K_{5}$	$K_{4}$
$c_{M+}$	$K_{4}$	$K_{4}$	$K_{4}$	$K_{5}$	$K_{5}$	$K_{5}$
$c_{M-}$	$K_{5}$	$K_{5}$	$K_{5}$	$K_{4}$	$K_{4}$	$K_{4}$
$v_{W}$	$K_{4}$	$K_{5}$	$K_{6}$	$K_{6}$	$K_{5}$	$K_{4}$
$v_{M}$	$K_{4}$	$K_{4}$	$K_{4}$	$K_{5}$	$K_{5}$	$K_{5}$
$v_{+}$	$K_{6}$	$K_{6}$	$K_{5}$	$K_{6}$	$K_{4}$	$K_{6}$
$v_{-}$	$K_{6}$	$K_{4}$	$K_{6}$	$K_{5}$	$K_{6}$	$K_{6}$

**Table 3 nanomaterials-13-01187-t003:** Calculated values of $S_{z}$ and $L_{z}$ for the relevant energy bands in monolayers.

	MoSe${}_{2}$		WSe${}_{2}$	
	$S_{z}$	$L_{z}$	$S_{z}$	$L_{z}$
$c_{+}$	−0.99	1.53	0.99	2.98
$c_{-}$	1.00	1.81	−0.89	1.88
$v_{+}$	1.00	3.94	1.00	5.02
$v_{-}$	−0.99	3.67	−0.98	4.07
$v_{\Gamma }$	0.96	0.06	0.87	0.19
$c_{Q+}$	−1.00	−0.09	−0.99	0.32
$c_{Q-}$	1.00	−0.16	0.96	0.05

**Table 4 nanomaterials-13-01187-t004:** Calculated values of $S_{z}$ and $L_{z}$ for the relevant energy bands in R- and H- type stackings.

	R${}_{h}^{h}$		R${}_{h}^{M}$		R${}_{h}^{X}$		H${}_{h}^{h}$		H${}_{h}^{M}$		H${}_{h}^{X}$	
	$S_{z}$	$L_{z}$	$S_{z}$	$L_{z}$	$S_{z}$	$L_{z}$	$S_{z}$	$L_{z}$	$S_{z}$	$L_{z}$	$S_{z}$	$L_{z}$
$c_{W+}$	0.97	2.98	0.97	3.02	0.97	2.97	0.97	2.92	0.97	2.98	0.97	3.02
$c_{W-}$	−0.88	1.88	−0.87	1.93	−0.88	1.88	−0.85	1.86	−0.89	1.89	−0.88	1.90
$c_{M+}$	−0.98	1.53	−0.98	1.55	−0.98	1.54	0.98	−1.54	0.98	−1.52	0.98	−1.56
$c_{M-}$	1.00	1.81	0.99	1.84	1.00	1.81	−1.00	−1.80	−1.00	−1.80	−1.00	−1.81
$v_{W}$	1.00	5.02	1.00	4.99	1.00	4.98	1.00	4.79	1.00	5.01	1.00	5.02
$v_{M}$	1.00	3.94	0.99	3.94	1.00	3.91	−1.00	−3.25	−1.00	−3.93	−1.00	−3.95
$v_{+}$	−0.98	4.04	−0.99	3.63	−0.98	4.01	0.99	−3.45	−0.98	4.08	0.24	−0.67
$v_{-}$	−0.99	3.70	−0.96	4.02	−0.99	3.62	−0.98	3.41	0.99	−3.66	−0.22	1.06
$v_{\Gamma }$	0.92	0.09	0.95	0.04	0.95	0.05	0.95	0.05	0.93	0.08	0.95	0.05
$c_{Q+}$	−0.93	0.11	−0.91	0.23	−0.87	0.14	−0.80	0.22	−0.59	0.12	−0.69	0.23
$c_{Q-}$	0.95	0.06	0.92	0.19	0.92	0.09	0.79	0.22	0.60	0.17	0.68	0.23

**Table 5 nanomaterials-13-01187-t005:** Symmetry-based selection rules for the direct interlayer excitons at the K point in the $C_{3}$ symmetry group.

	$v_{W}\left(K\right)\to c_{M-}\left(K\right)$	$v_{W}\left(K\right)\to c_{M+}\left(K\right)$
R${}_{h}^{h}$	$K_{4}^{*}⊗K_{5}=K_{3}\Rightarrow \sigma_{+}$	$K_{4}^{*}⊗K_{4}=K_{1}\Rightarrow z$
R${}_{h}^{M}$	$K_{5}^{*}⊗K_{5}=K_{1}\Rightarrow z$	$K_{5}^{*}⊗K_{4}=K_{2}\Rightarrow \sigma_{-}$
R${}_{h}^{X}$	$K_{6}^{*}⊗K_{5}=K_{2}\Rightarrow \sigma_{-}$	$K_{6}^{*}⊗K_{4}=K_{3}\Rightarrow \sigma_{+}$
H${}_{h}^{h}$	$K_{6}^{*}⊗K_{4}=K_{3}\Rightarrow \sigma_{+}$	$K_{6}^{*}⊗K_{5}=K_{2}\Rightarrow \sigma_{-}$
H${}_{h}^{M}$	$K_{5}^{*}⊗K_{4}=K_{2}\Rightarrow \sigma_{-}$	$K_{5}^{*}⊗K_{5}=K_{1}\Rightarrow z$
H${}_{h}^{X}$	$K_{4}^{*}⊗K_{4}=K_{1}\Rightarrow z$	$K_{4}^{*}⊗K_{5}=K_{3}\Rightarrow \sigma_{+}$

**Table 6 nanomaterials-13-01187-t006:** Direct product of valence bands at the −K point and conduction bands at the K point in the $C_{3}$ symmetry group.

	$v_{W}\left(-K\right)\to c_{M-}\left(K\right)$	$v_{W}\left(-K\right)\to c_{M+}\left(K\right)$
R${}_{h}^{h}$	$K_{5}^{*}⊗K_{5}=K_{1}$	$K_{5}^{*}⊗K_{4}=K_{2}$
R${}_{h}^{M}$	$K_{4}^{*}⊗K_{5}=K_{3}$	$K_{4}^{*}⊗K_{4}=K_{1}$
R${}_{h}^{X}$	$K_{6}^{*}⊗K_{5}=K_{2}$	$K_{6}^{*}⊗K_{4}=K_{3}$
H${}_{h}^{h}$	$K_{6}^{*}⊗K_{4}=K_{3}$	$K_{6}^{*}⊗K_{5}=K_{2}$
H${}_{h}^{M}$	$K_{4}^{*}⊗K_{4}=K_{1}$	$K_{4}^{*}⊗K_{5}=K_{3}$
H${}_{h}^{X}$	$K_{5}^{*}⊗K_{4}=K_{2}$	$K_{5}^{*}⊗K_{5}=K_{1}$

**Table 7 nanomaterials-13-01187-t007:** Direct product of valence bands at the Γ point and conduction bands at the K point in the $C_{3}$ symmetry group.

	R${}_{h}^{h}$, R${}_{h}^{M}$, R${}_{h}^{X}$	H${}_{h}^{h}$, H${}_{h}^{M}$, H${}_{h}^{X}$
$v_{\Gamma }\left(K_{4}\right)\to c_{M-}\left(K\right)$	$K_{4}^{*}⊗K_{5}=K_{3}$	$K_{4}^{*}⊗K_{4}=K_{1}$
$v_{\Gamma }\left(K_{5}\right)\to c_{M-}\left(K\right)$	$K_{5}^{*}⊗K_{5}=K_{1}$	$K_{5}^{*}⊗K_{4}=K_{2}$
$v_{\Gamma }\left(K_{4}\right)\to c_{M+}\left(K\right)$	$K_{4}^{*}⊗K_{4}=K_{1}$	$K_{4}^{*}⊗K_{5}=K_{3}$
$v_{\Gamma }\left(K_{5}\right)\to c_{M+}\left(K\right)$	$K_{5}^{*}⊗K_{4}=K_{2}$	$K_{5}^{*}⊗K_{5}=K_{1}$

**Table 8 nanomaterials-13-01187-t008:** Dipolar exciton g-factors involving the top valence band states (at Γ, K, and −K points) to conduction bands of Mo at the K point ($c_{M\pm }$) or to the conduction bands at the Q point ($c_{Q\pm }$) at zero electric field and equilibrium interlayer distance. The g-factors with unambiguously determined signs are given in bold.

	R${}_{h}^{h}$	R${}_{h}^{M}$	R${}_{h}^{X}$	H${}_{h}^{h}$	H${}_{h}^{M}$	H${}_{h}^{X}$
$v_{W}\left(K\right)\to c_{M-}\left(K\right)$	−6.42	6.34	+6.34	−17.18	+17.62	17.66
$v_{W}\left(K\right)\to c_{M+}\left(K\right)$	10.93	+10.85	−10.84	+12.70	13.10	−13.18
$v_{W}\left(K\right)\to c_{Q-}\left(Q\right)$	10.01	9.78	9.94	9.56	10.49	10.22
$v_{W}\left(K\right)\to c_{Q+}\left(Q\right)$	13.67	13.34	13.42	12.74	12.94	12.94
$v_{W}\left(-K\right)\to c_{M-}\left(K\right)$	17.64	17.63	17.57	5.99	6.43	6.41
$v_{W}\left(-K\right)\to c_{M+}\left(K\right)$	13.13	13.12	13.07	10.47	10.94	10.89
$v_{W}\left(-K\right)\to c_{Q-}\left(Q\right)$	14.05	14.19	13.97	13.61	13.55	13.85
$v_{W}\left(-K\right)\to c_{Q+}\left(Q\right)$	10.39	10.63	10.49	10.43	11.10	11.12
$v_{\Gamma_{1}}\to c_{M-}\left(K\right)$	3.58	3.67	3.62	7.58	7.62	7.62
$v_{\Gamma_{1}}\to c_{M+}\left(K\right)$	0.92	0.85	0.87	3.10	3.10	3.14
$v_{\Gamma_{1}}\to c_{Q-}\left(Q\right)$	0.01	0.23	0.03	0.03	0.49	0.18
$v_{\Gamma_{1}}\to c_{Q+}\left(Q\right)$	3.66	3.33	3.46	3.14	2.94	2.90
$v_{\Gamma_{2}}\to c_{M-}\left(K\right)$	7.63	7.63	7.60	3.60	3.58	3.63
$v_{\Gamma_{2}}\to c_{M+}\left(K\right)$	3.13	3.12	3.11	0.88	0.94	0.85
$v_{\Gamma_{2}}\to c_{Q-}\left(Q\right)$	4.04	4.19	4.01	4.01	3.55	3.81
$v_{\Gamma_{2}}\to c_{Q+}\left(Q\right)$	0.39	0.63	0.52	0.83	1.10	1.08

## Data Availability

Not applicable.

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
