# Peer review of "Signatures of Electric Field and Layer Separation Effects on the Spin-Valley Physics of MoSe2/WSe2 Heterobilayers: From Energy Bands to Dipolar Excitons"

_nanomaterials, 2023, doi:10.3390/nano13071187_

Round 1

Reviewer 1 Report

In ”Signatures of electric field and layer separation effects on the spin-valley physics of MoSe2/WSe2 heterobilayers: from energy bands to dipolar excitons,” the authors theoretically investigate electric field and interlayer distance dependence of interlayer exciton optical properties. They investigate both R and H type MoSe2-WSe2 heterostructures and show that H type heterostructures should exhibit strongly varying excitonic g factors as a function of out of plane electric field. They explain that this is due to a stronger dependence of band mixing on applied electric field in H type structures.

Overall, the paper is well written and organized, and the results appear to be reasonable. The paper is acceptable for publication as is, however, it could be significantly improved by making the changes suggested below.

1)      The electric fields currently range from -1 to +2 volts. Why is the range not symmetric?

2)      It would help an experimentalist if the authors plot theoretical (co and cross polarized) PL emission (intensity+energy) as a function of electric field.   This would help the reader visualize the results instead of trying to piece together the optical dipole moment and individual band energies.

3)      Can the authors calculate the interlayer exciton lifetime as a function of electric field based on their optical dipole moment calculations? If so, that would be very helpful for an experimental reader to compare to their results.

Reviewer 2 Report

The manuscript presents a first-principles computational investigation of the band structures, spin and orbital angular momenta, g-factors and optical matrix elements of the Bloch states of MoSe2/WSe2 hetero-bilayer under external fields. The first principles studies provide a useful ingredient for gaining deep insight into the spin-valley properties and microscopic structures of the Bloch states of the hetero-bilayer systems. The technical methodology is not trivial, especially in the orbital angular momentum part which requires hundreds of bands to achieve high accuracy of the numerical results. The manuscript is also well organized and provides a useful database of material information for the 2D systems. I recommend the manuscript be published in the journal "Nanomaterials".
I have only a question about the studies.

The work basically analyzes only the quasi-particle Bloch states and accordingly infers the spin properties such as spin angular momentum and g-factors of the exciton. An exciton is essentially a correlated excited state that superpositions many single particles transitions rather than a single Bloch state transition. About the spin mechanisms, besides the intrinsic spin-orbit interaction that is considered in the first principles studies of the work, the electron-hole exchange interaction and field-driven Rashba spin-orbit interaction could also affect the spin properties of the exciton. Can the authors justify that the studies based on the quasi-particle band structure are applicable to infer the spin properties of exciton?
